# The Influence of Short-Term Weather Parameters and Air Pollution on Adolescent Airway Inflammation

**DOI:** 10.3390/ijerph20196827

**Published:** 2023-09-25

**Authors:** Ashtyn Tracey Areal, Nidhi Singh, Qi Zhao, Dietrich Berdel, Sibylle Koletzko, Andrea von Berg, Monika Gappa, Joachim Heinrich, Marie Standl, Michael J. Abramson, Tamara Schikowski

**Affiliations:** 1IUF—Leibniz Research Institute for Environmental Medicine, 40225 Düsseldorf, Germany; ashtyntracey.areal@iuf-duesseldorf.de (A.T.A.); nidhi.singh@iuf-duesseldorf.de (N.S.); qi.zhao@sdu.edu.cn (Q.Z.); 2Department of Epidemiology, Medical Research School, Heinrich-Heine-University, 40225 Düsseldorf, Germany; 3School of Public Health, Cheeloo College of Medicine, Shandong University, Jinan 250100, China; 4Department of Pediatrics, Research Institute, Marien-Hospital Wesel, 46483 Wesel, Germany; berdel.vonberg@t-online.de (D.B.); avb.rodehorst@gmx.de (A.v.B.); 5Department of Pediatrics, Dr. von Hauner Children’s Hospital Munich, University Hospital, LMU Munich, 80539 Munich, Germany; sibylle.koletzko@med.uni-muenchen.de; 6Department of Pediatrics, Gastroenterology and Nutrition, School of Medicine Collegium Medicum, University of Warmia and Mazury, 10-082 Olsztyn, Poland; 7Department of Paediatrics, Evangelisches Krankenhaus, 40217 Düsseldorf, Germany; monika.gappa@evk-duesseldorf.de; 8Institute and Clinic for Occupational, Social and Environmental Medicine, University Hospital, LMU Munich, 80539 Munich, Germany; joachim.heinrich@med.uni-muenchen.de; 9German Center for Lung Research (DZL), 35392 Gießen, Germany; marie.standl@helmholtz-muenchen.de; 10Allergy and Lung Health Unit, Melbourne School of Population and Global Health, The University of Melbourne, Melbourne, VIC 3010, Australia; 11Institute of Epidemiology, Helmholtz Zentrum München—German Research Center for Environmental Health, 85764 Neuherberg, Germany; 12School of Public Health & Preventive Medicine, Monash University, Melbourne, VIC 3004, Australia; michael.abramson@monash.edu

**Keywords:** relative humidity, air pollution, environmental epidemiology, fraction of exhaled nitric oxide, adolescent, cohort studies

## Abstract

Fraction of exhaled Nitric Oxide (FeNO) is a marker of airway inflammation. We examined the main effects and interactions of relative humidity (RH) and air pollution on adolescents’ FeNO. Two thousand and forty-two participants from the 15-year follow-up of the German GINIplus and LISA birth cohorts were included. Daily meteorological (maximum [Tmax], minimum [Tmin] and mean [Tmean] temperatures and RH) and air pollution [Ozone (O_3_), nitrogen dioxide (NO_2_) and particulate matter < 2.5 µm (PM_2.5_)] were assessed. Linear models were fitted with Ln(FeNO) as the outcome. Increases in FeNO indicate an increase in lung inflammation. Increased FeNO was associated with an increase in temperature, PM_2.5_, O_3_ and NO_2_. A 5% increase in RH was associated with a decrease in FeNO. Interactions between RH and high (*p* = 0.007) and medium (*p* = 0.050) NO_2_ were associated with increases in FeNO; while interactions between RH and high (*p* = 0.042) and medium (*p* = 0.040) O_3_ were associated with decreases in FeNO. Adverse effects were present for male participants, participants with low SES, participants with chronic respiratory disease, and participants from Wesel. Short-term weather and air pollution have an effect on lung inflammation in German adolescents. Future research should focus on further assessing the short-term effect of multiple exposures on lung inflammation in adolescents.

## 1. Introduction

The absolute global burden of chronic respiratory diseases has increased since 1990 [1]. Whilst tobacco smoking remains the leading cause of respiratory disability in men, household and ambient air pollution are the predominant risk factors for women in many regions of the world. Globally, in 2017, ambient ozone (O_3_) and particulate matter (PM) pollution were associated with 96.4 and 206 Disability Adjusted Life Years (DALYs) per 100,000 people of all ages, respectively [1]. As climate change accelerates, there is increasing interest in the relationship between weather variables and respiratory health outcomes. Relative humidity (RH) and temperature have typically been treated as confounders in time-series studies of air pollution and all-cause or respiratory mortality [2].

During periods of increased lung inflammation, the concentration of nitric oxide accumulates in the lungs and can be measured during exhalation [3]. Fractional exhaled nitric oxide (FeNO) is a noninvasive biomarker that assesses lung inflammation and assists in the diagnosis and assessment of asthma [4]. Previous studies in children, young adults (aged 20 and above) and the elderly found that exposure to O_3_, PM with a diameter less than 2.5 µm (PM_2.5_), PM with a diameter less than 10 µm (PM_10_) and nitrogen dioxide (NO_2_), and ambient temperature were associated with an increase in FeNO [4,5,6,7,8,9].

Although the short-term dose–response relationship between air pollution and FeNO has been described in children, young and older adults, there are limited data for adolescents. Additionally, there are no studies that provide information on the effect of RH on FeNO. This is concerning as adolescence is an important period of lung development as physical growth is rapid, and asthma becomes more common in females than in males [10].

Information on the interactive or modifying effect of weather and air pollution on FeNO is limited and, as far as we are aware, no research looking into how this interaction impacts adolescents during a crucial time of growth. Thus, this analysis aimed to examine the main effects and interactions of low-level short-term air pollutants and weather variables on adolescents’ airway inflammation (FeNO).

## 2. Materials and Methods

### 2.1. Study Population

Participants were recruited for two ongoing German population-based, birth cohort studies, which recruited healthy full-term neonates with normal birthweight in Munich and Wesel. The German Infant Study on the Influence of Nutrition Intervention plus Air Pollution and Genetics on Allergy Development (GINIplus) recruited a total of 5991 neonates in Munich and Wesel between September 1995 and July 1998. The Influence of Lifestyle Factors on the Development of the Immune System and Allergies in East and West Germany Study (LISA) recruited a total of 3097 neonates in Bad Honnef, Leipzig, Munich and Wesel between November 1997 and January 1999. The study areas of the cohorts are shown in Appendix A. Data from these two birth cohorts were collected at birth as well as three follow-ups, which occurred at ages 6, 10, and 15, and then due to their harmonised design, pooled for Wesel (GINIplus/LISA North: number = 3390) and Munich (GINIplus/LISA South: number = 4413). Parents completed questionnaires that collected data on respiratory conditions and covariates such as the sex of the child, parental/personal smoking and socioeconomic status (parental education). Further details of recruitment and follow-up to 15 years have been presented elsewhere [11]. The data in this analysis were from the 15-year follow-up assessments for both cohorts in Munich and Wesel. Ethical approval was granted by the Bavarian Board of Physicians (10090 and 12067), Board of Physicians of North-Rhine Westphalia (20101424 and 2012446), and Board of Physicians of Saxony (EK-BR-02/13-1). The parents of participants provided written informed consent.

### 2.2. Assessment of Lung Function

Fraction of exhaled nitric oxide (FeNO) is a well-established biomarker of airway inflammation. FeNO is routinely used in clinical practice in many countries and has also been investigated as a biomarker in epidemiological studies of air pollution. FeNO measurements, which were adjusted for the nonlinear effects of age, height, weight, and sex, were made between the years 2011 and 2013 with a handheld device (NIOX MINO, Aerocrine, Solna, Sweden) following the guidelines of the American Thoracic Society and European Respiratory Society [12]. Any respiratory tract infections, personal smoking and anti-inflammatory medications were recorded. Since FeNO followed an approximately log-normal distribution, the data were loge-transformed before analysis.

### 2.3. Assessment of Environmental Exposures

Short-term air pollution exposure was assessed as average concentrations of 24 h O_3_, NO_2_ and PM_2.5_. The air pollutant exposures at participants’ 15-year residential addresses were estimated at a spatial resolution of 2 × 2 km by chemical transport models and data provided by the German Environment Agency (Umwelt Bundesamt, UBA [13]). Weather variables (daily maximum (Tmax), minimum (Tmin) and mean (Tmean) temperature and RH) were obtained for Munich and Wesel from the German Weather Service’s high-resolution reanalysis system COSMO-REA6 at a spatial resolution of 6 × 6 km [14]. The warm season was defined as May to October and the cold season as November to April.

### 2.4. Statistical Analysis

There was little variation in temperature and RH between the study sites in Munich and Wesel; as such, it was decided to pool the participants to increase the power of the statistical analysis as in previous studies [15]. We performed correlation tests and checked the collinearity between variables as well as normality tests. Linear regression models were fitted with Ln(FeNO) as the outcome and continuous RH as our main exposure. We determined the main effects for continuous RH and air pollution (i.e., O_3_, NO_2_ and PM_2.5_). The model was further adjusted for age, height, weight, sex, a temperature variable (Tmax, Tmin, or Tmean), season with the warm season as the reference category, history of respiratory disease with “No” as the reference category, and location with Munich as the reference category. An interaction model was then fitted with an interaction term between RH and categorical air pollution included in the main model. The air pollution categorical variable was defined as the following: <25% (Low), 25–75% (Medium), and >75% (High). We chose “Low” as the reference category, as it represented the optimum exposure, while “Medium” represented the most common exposure and “High” represented nonoptimum exposure. Effect modification was examined by the site (Munich/Wesel), binary sex characterisation (female/male), maximum parental education as an indicator of socioeconomic status (SES), body mass index (BMI) and history of respiratory conditions. Respiratory conditions were defined as a history of asthma, a history of chronic bronchitis, a history of chronic wheeze, and/or asthma, chronic bronchitis and/or wheeze at the time of assessment.

### 2.5. Sensitivity Analysis

To test the robustness of the core model, sensitivity analyses were conducted to explore the lagged effects up to 10 days prior, location, age, the effect of sex, history of respiratory conditions, height, weight, maximal parental education (as an indicator of SES) and parental smoking. Based on the sensitivity analysis, we chose the Lag01 (one-day moving average) effect for all environmental factors in the core model.

Statistical analysis and data summary were conducted in R version 4.0.4 (15 February 2021) using the packages “stats”, “gtsummary”,“MASS“, and “data.table” [16]. A decrease in FeNO indicates a decrease in lung inflammation. Results were calculated as a percentage increase per 5% increase in RH. A two-sided *p*-value < 0.05 was considered statistically significant.

## 3. Results

### 3.1. Description of Participants and Exposures

This analysis included 2042 participants, 1191 participants in Munich and 851 in Wesel. Their mean (±Standard Deviation [SD]) age was 15.06 (±0.29) years. There were slightly more females than males (51% vs. 49%) (Table 1). The majority of the parents in Munich than in Wesel had completed over 10 years of education (78.3% vs. 52.6%). Just under a third of participants in both centres had a history of respiratory conditions. Approximately 70% of participants had a normal BMI, while 21.58% of participants were underweight and 9.7% were overweight. More participants from Wesel were overweight than participants from Munich (12.10% vs. 7.98%) (Table 1).

Tmin, Tmean, Tmax, and RH were similar in Munich and Wesel (Table 2). Among air pollutants, PM_2.5_ and O_3_ concentrations were slightly higher in Wesel, while NO_2_ concentrations were higher in Munich. Most measurements of FeNO were performed during the warm season.

### 3.2. The Main Effects of Weather Variables and Air Pollution on FeNO

A 5% increase in RH showed a consistent nonsignificant trend towards a decrease in FeNO, and, as such, a decrease in lung inflammation across all temperature and air pollution models (percentage change = −0.01%; 95% CI: −0.03 to 0.01) (Table 3). PM_2.5_ (percentage change = 0.19; 95% CI: −0.23 to 0–64), O_3_ (percentage change = 0.02; 95% CI: −0.18 to 0.20), and NO_2_ (percentage change = 0.31; 95% CI: −0.08 to 0.71) were all associated with an increase in FeNO and therefore an increase in lung inflammation; however, this result was not statistically significant (Table 3). Tmax, Tmin, and Tmean were all not significantly associated with an increase in FeNO (Table 3).

### 3.3. Interactive Effects of RH and Air Pollution on FeNO

The interactive effect between RH and PM_2.5_ showed a nonsignificant trend towards an increase in FeNO for days with medium (percentage change = 0.02; 95% CI: −0.02 to 0.05) and high (percentage change = 0.02; 95% CI: −0.02 to 0.06) PM_2.5_ concentrations compared to days with low PM_2.5_ concentrations (Table 4). There was a statistically significant decrease in FeNO per 5% increase in RH on days with medium (percentage change = −0.04; 95% CI: −0.08 to 0.00) and high (percentage change = −0.04; 95% CI: −0.09 to 0.00) O_3_ concentrations (Table 5). On days with medium (percentage change = 0.03; 95% CI: 0.00 to 0.07) and high (percentage change = 0.05; 95% CI: 0.01 to 0.08) concentrations of NO_2_, there was a statistically significant increase in FeNO per 5% increase in RH (Table 6).

### 3.4. Effect Modification

When we stratified the analysis by sex, we found that RH was associated with a statistically nonsignificant decrease in FeNO in both male and female participants (Appendix A). PM_2.5_ and NO_2_ both showed a nonsignificant trend towards an increase in FeNO in both male and female participants. However, O_3_ was associated with a decrease in FeNO in female participants, while in male participants, O_3_ was associated with an increase in FeNO (Appendix A). An increase in temperature was associated with an increase in FeNO in both male and female participants; however this effect, while not statistically significant, was stronger in male participants than female participants (Appendix A).

When we assessed the modifying effect of BMI, we found that an increase in RH was consistently associated with a decrease in FeNO (Appendix A). PM_2.5_ was associated with a decrease in FeNO in underweight and overweight participants, while in participants who were classified as having normal weight, PM_2.5_ was associated with a decrease in FeNO (Appendix A). O_3_ was associated with a decrease in FeNO in both underweight and normal-weight participants, while in overweight participants, O_3_ was associated with an increase in FeNO (Appendix A). NO_2_ was consistently associated with an increase in FeNO across all participants (Appendix A). Temperature increases were associated with an increase in FeNO across all participants; however, this effect was stronger in overweight participants (Appendix A).

RH was associated with an increase in FeNO per 5% increase in RH in low SES participants, while in medium and high SES participants, RH was associated with a decrease in FeNO (Appendix A). NO_2_ was consistently associated with an increase in FeNO for all participants; however, this effect was stronger in low SES participants (Appendix A). PM_2.5_ was associated with an increase in FeNO in low and high SES participants, while in medium SES participants, PM_2.5_ was associated with a decrease in FeNO (Appendix A). O_3_ was associated with a decrease in FeNO in both low and high SES participants; however, O_3_ was associated with an increase in FeNO in those with medium SES (Appendix A). Temperature was associated with an increase in FeNO across all participants; however, the effect was stronger in low SES participants (Appendix A).

In those with CRD, RH and NO_2_ showed a nonsignificant trend towards a decrease in FeNO; while PM_2.5_, O_3_, and temperature were associated with an increase in FeNO (Appendix A). In those without CRD, RH, PM_2.5_, and temperature were associated with a nonsignificant trend towards an increase in FeNO, and NO_2_ was statistically significantly associated with an increase in FeNO (Appendix A). O_3_ was associated with a decrease in FeNO in those without CRD (Appendix A). The effect of temperature was greater in those with CRD.

In participants from Wesel, RH, O_3_, NO_2_, and temperature all showed nonsignificant trends towards an increase in FeNO, while PM_2.5_ was significantly associated with an increase in FeNO (Appendix A). RH, PM_2.5_, and O_3_ were all associated with a decrease in FeNO in participants from Munich, while NO_2_ and temperature were both associated with an increase in FeNO (Appendix A).

### 3.5. Sensitivity Analyses

We conducted a series of sensitivity analyses to test the robustness of the results. First, we used different lags of RH, temperature and air pollution for up to 10 days. While results were similar across all lag periods for RH, temperature, and all air pollutants, it was found that these exposures were most associated with FeNO at Lag01. Secondly, we adjusted the model for additional covariates. However, associations of prior day RH with FeNO were unchanged after adjusting for age, height, weight, sex, respiratory tract infections, personal and second-hand smoking, family history of respiratory disease, and anti-inflammatory medications.

## 4. Discussion

This analysis of a large cohort of 15-year-old German adolescents has shown that FeNO, a marker of airway inflammation, was consistently associated with short-term RH, temperature, air pollution and interactions between RH and air pollution. There were no statistically significant main effects; however, important trends were apparent. Increases in air pollution and temperature were both associated with an increase in lung inflammation, while increases in RH were associated with a decrease in lung inflammation. Interactions between RH and PM_2.5_ indicated a nonsignificant trend towards an increase in FeNO per 5% increase in RH on days with medium (25th to 75th percentile) and high (>75th percentile) daily average concentrations compared to days with low (<25th percentile) concentrations. There were significant associations between RH and O_3_, and RH and NO_2_; there was a significant increase in FeNO per 5% increase in RH on medium and high NO_2_ concentration days compared to low concentration days. On days with medium and high O_3_ concentrations, there was a decrease in FeNO per 5% increase in RH, this could be because RH could counter the adverse effects of O_3_.

When we stratified the analysis by sex, we found that, while not statistically significant, male participants experienced a stronger effect of temperature than female participants. Participants with a low SES were more likely to experience adverse effects of RH, NO_2_, and temperature than those with a higher SES. Participants with CRD experienced an increase in lung inflammation with increasing RH, temperature and O_3_ concentrations. Participants from Wesel were more likely than participants from Munich to experience an increase in FeNO with increasing RH, temperature, O_3_, PM_2.5_, and NO_2_.

Germany generally has a temperate rainy climate with high levels of humidity and consistently moderate temperatures [17,18]. However, there were different sources of air pollution, with high traffic-related emissions likely explaining the higher concentrations of NO_2_ in urban Munich compared to rural Wesel. On the other hand, agricultural emissions are a major source of PM_2.5_ in rural Wesel compared to urban Munich [19].

It is not that straightforward to put our results in the context of previous research for several reasons. Weather variables have typically been regarded only as confounders in respiratory epidemiology and not much has been published on associations with markers such as FeNO. Most research has concentrated on long-term exposure to environmental factors. Considering that FeNO is sensitive to external factors, investigating how short-term exposure impacts FeNO is of interest [3]. While we failed to find any significant associations between FeNO and PM_2.5_ in our adolescent participants, several studies investigating the effect of PM_2.5_ on university students did find adverse effects. A panel study of university students in a highly polluted city in China (72 < weekly mean PM_2.5_ < 180 µg/m^3^) found that temperature and PM_2.5_ were both positively associated with FeNO [4]. Also, a study of healthy university students exercising in a highly polluted city in Poland (median indoor PM_2.5_ 114 vs. 26.5 µg/m^3^) found that increased FeNO during high exposure was associated with higher outdoor PM_10_, NO_2_ and RH [5].

More is known about the long-term effects of air pollution on airway inflammation. We have previously shown that long-term exposures to NO_2_, PM_2.5_ and PM_10_ were associated with increased FeNO in a cohort of older (mean age 75) German women [6,7]. While the Southern California Children’s Health Study recently reported that long-term (annual) exposures to PM_2.5_ and NO_2_ were associated with increased FeNO after adjustment for covariates, including sex, asthma, second-hand tobacco smoke, temperature and short-term pollutant exposures [8,9]. The limited information on short-term exposures highlights the necessity for this study, which helps to fill this gap in the literature.

The findings of our analysis have biological plausibility. We found that Tmax, Tmin, and Tmean were associated with an increase in FeNO. This is consistent with literature that found that cold seasons and low temperatures have long been associated with respiratory infections and exacerbation of respiratory conditions, probably because people congregate more indoors [20]. Indeed, due to cold dry air being a common asthma trigger, the cold dry air challenge is used as a diagnostic test for asthma in children [21]. On the other hand, children with asthma have often been encouraged to take up swimming, because the warm moist air does not trigger attacks, in contrast to other sports such as running or cycling [22].

Females typically have a lower metabolic rate, lower skin temperature, lower body mass, higher body fat, and less surface area, but a higher surface area to mass ratio than males. Additionally, females have a slower blood flow, indicating that females are more sensitive to low temperatures than males, who are more sensitive to high temperatures, as cold exposure causes their skin temperature to lower even further, especially in the extremities [23,24].

To obtain further insights into the causal pathways, it is necessary to study surrogate subclinical endpoints such as lung function and biomarkers. Further investigation includes investigating epigenetic markers, as 15 genes have been identified whose methylation status is associated with ambient temperature [25]. Further studies should also be conducted examining other systemic inflammatory biomarkers such as blood neutrophil and eosinophil counts, serum interleukin 6 [26], C reactive protein [27], etc.

This analysis has several strengths. The data were obtained from well-characterised birth cohorts. Short-term air pollution and meteorological exposures were estimated by well-validated high-resolution models. An objective marker of airway inflammation was measured following standard guidelines [12].

However, there were also some limitations: Participant numbers were low, limiting statistical power in some analyses. Although the GINIplus/LISA cohort has been well described, the findings might not be generalisable to adolescents in other countries with higher levels of air pollution and/or different meteorological conditions.

## 5. Conclusions

This analysis of a large data set of German adolescents from two birth cohorts demonstrates that there is an interaction between climate variables and air pollution and FeNO, which is supported by those observed in other age groups. An increase in lung inflammation was associated with the interacting effects of RH and air pollution in this cohort. These findings may have important clinical implications, as they indicate an increase in negative respiratory health outcomes and provide evidence on a relatively unknown topic. Considering the acceleration of climate change, future research should focus further not just on the potential impacts of extreme climate events or individual exposure effects on health, but also on the short- and long-term impacts of daily weather variables as well as the effect of multiple exposures on all facets of health.

## Figures and Tables

**Table 1 ijerph-20-06827-t001:** Description of participants.

Characteristic	Overall ^1^	Munich ^1^	Wesel ^1^	*p*-Value ^2^
Number of Participants	2042	1191	851	
Age	15.06 (0.29)	15.09 (0.29)	15.02 (0.28)	0.049
Sex Female Male	1050 (51%) 992 (49%)	609 (51%) 582 (49%)	441 (52%) 410 (48%)	0.8
Height (cm)	171.4 (8.30)	170.8 (8.22)	172.4 (8.32)	<0.001
Weight (Kg)	61.74 (11.96)	60.46 (11.10)	63.53 (12.86)	<0.001
FeNO (ppb)	23.1 (20.94)	25.48 (22.60)	19.77 (17.86)	<0.001
Respiratory condition Yes No NA	643 (31.5%) 1397 (68.4%) 2 (0.1%)	388 (32.58%) 801 (67.25%) 2 (0.17%)	255 (29.96%) 596 (70.04%) 0	0.2
Maximal parental education Low (<10 years) Medium (=10 years) High (>10 years) NA	118 (5.78%) 539 (26.40%) 1380 (67.58%) 5 (0.24%)	46 (3.86%) 210 (17.63%) 932 (78.25%) 3 (0.25%)	72 (8.46%) 329 (38.66%) 448 (52.64%) 2 (0.24%)	<0.001
Body Mass Index (Kg/m^2^) Low (<18.5) Normal (18.5–24.9) High (>25)	414 (20.27%) 1430 (70.03%) 198 (9.70%)	257 (21.58%) 839 (70.45%) 95 (7.98%)	157 (18.45%) 591 (69.45%) 103 (12.10%)	0.004

^1^ Mean (Standard Deviation); n (%). ^2^ Wilcoxon rank sum test; Pearson’s Chi-squared test.

**Table 2 ijerph-20-06827-t002:** Description of environmental factors.

Characteristic	Overall ^1^	Munich ^1^	Wesel ^1^	*p*-Value ^2^
Season Warm Cold	1315 (64%) 727 (36%)	776 (65%) 415 (35%)	539 (63%) 312 (37%)	0.4
Relative Humidity (%)	75.22 (10.60)	75.03 (11.07)	75.49 (9.90)	0.047
Tmax (°C)	16.43 (7.83)	16.50 (8.24)	16.34 (7.22)	0.6
Tmin (°C)	8.61 (6.08)	8.36 (6.13)	8.95 (6.00)	0.058
Tmean (°C)	12.36 (6.80)	12.24 (7.00)	12.53 (6.50)	0.4
PM_2.5_ (µg/m^3^)	11.09 (6.58)	9.75 (6.34)	12.96 (6.46)	<0.001
NO_2_ (µg/m^3^)	13.17 (8.39)	16.29 (8.79)	8.79 (5.28)	<0.001
O_3_ (µg/m^3^)	53.54 (19.76)	49.96 (20.98)	58.55 (16.68)	<0.001

^1^ Mean (standard deviation); n (%). ^2^ Wilcoxon rank sum test; Pearson’s Chi-squared test.

**Table 3 ijerph-20-06827-t003:** The main effects of RH, air pollution, and temperature on FeNO in a cohort of German adolescents.

	Tmax		Tmin		Tmean	
	Percentage Change (95% CI) *	*p*-Value ^1,^*	Percentage Change (95% CI) *	*p*-Value ^1,^*	Percentage Change (95% CI) *	*p*-Value ^1,^*
RH ^2^	−0.01 (−0.02, 0.01)	0.265	−0.01 (−0.03, 0.00)	0.074	−0.01 (−0.02, 0.00)	0.179
PM_2.5_	0.19 (−0.24, 0.63)	0.382	0.20 (−0.23, 0.64)	0.355	0.20 (−0.23, 0.63)	0.367
Temperature	0.29 (−0.28, 0.86)	0.314	0.27 (−0.41, 0.94)	0.439	0.32 (−0.31, 0.96)	0.321
RH ^2^	−0.01 (−0.03, 0.01)	0.316	−0.01 (−0.03, 0.00)	0.144	−0.01 (−0.03, 0.01)	0.237
O_3_	0.01 (−0.18, 0.19)	0.949	0.02 (−0.17, 0.20)	0.849	0.01 (−0.18, 0.20)	0.928
Temperature	0.31 (−0.29, 0.91)	0.308	0.25 (−0.45, 0.95)	0.486	0.33 (−0.33, 1.00)	0.328
RH ^2^	−0.01 (−0.03, 0.01)	0.199	**−0.01 (−0.03, −0.00)**	**0.039**	−0.01 (−0.03, 0.00)	0.122
NO_2_	0.31 (−0.29, 0.91)	0.117	0.32 (−0.07, 0.71)	0.107	0.32 (−0.07, 0.71)	0.109
Temperature	0.35 (−0.22, 0.92)	0.225	0.35 (−0.33, 1.04)	0.311	0.40 (−0.24, 1.04)	0.222

^1^ *p*-value < 0.05 in bold. ^2^ per 5% increase in RH at Lag01. * Adjusted for indicated study location, season, chronic respiratory disease.

**Table 4 ijerph-20-06827-t004:** The interactive effects of RH and PM_2.5_ on FeNO in a cohort of German Adolescents.

	Interaction Term ^2,^*	Percentage Change (95% CI) ^2,^*	*p*-Value ^1,^*
Tmax *	RH: High PM_2.5_	0.02 (−0.02, 0.06)	0.256
	RH: Medium PM_2.5_	0.02 (−0.02, 0.05)	0.388
Tmin *	RH: High PM_2.5_	0.02 (−0.02, 0.06)	0.256
	RH: Medium PM_2.5_	0.02 (−0.02, 0.05)	0.389
Tmean *	RH: High PM_2.5_	0.02 (−0.02, 0.06)	0.258
	RH: Medium PM_2.5_	0.02 (−0.02, 0.05)	0.389

^1^ *p*-value < 0.05 in bold. ^2^ per 5% increase in RH at Lag01. * Adjusted for indicated study location, season, chronic respiratory disease, and indicated temperature.

**Table 5 ijerph-20-06827-t005:** The interactive effects of RH and O_3_ on FeNO in a cohort of German Adolescents.

	Interaction Term ^2,^*	Percentage Change (95% CI) ^2,^*	*p*-Value ^1,^*
Tmax *	RH: High O_3_	**−0.04 (−0.09, −0.00)**	**0.042**
	RH: Medium O_3_	**−0.04 (−0.07, −0.00)**	**0.040**
Tmin *	RH: High O_3_	**−0.04 (−0.09, −0.00)**	**0.042**
	RH: Medium O_3_	**−0.04 (−0.08, −0.00)**	**0.038**
Tmean *	RH: High O_3_	**−0.04 (−0.09, −0.00)**	**0.043**
	RH: Medium O_3_	**−0.04 (−0.07, −0.00)**	**0.040**

^1^ *p*-value < 0.05 in bold. ^2^ per 5% increase in RH at Lag01. * Adjusted for indicated study location, season, chronic respiratory disease, and indicated temperature.

**Table 6 ijerph-20-06827-t006:** The interactive effects of RH and NO_2_ on FeNO in a cohort of German Adolescents.

	Interaction Term ^2,^*	Percentage Change (95% CI) ^2,^*	*p*-Value ^1,^*
Tmax *	RH: High NO_2_	**0.05 (0.01, 0.08)**	**0.007**
	RH: Medium NO_2_	**0.03 (0.00, 0.07)**	**0.050**
Tmin *	RH: High NO_2_	**0.05 (0.01, 0.08)**	**0.008**
	RH: Medium NO_2_	**0.03 (0.00, 0.07)**	**0.050**
Tmean *	RH: High NO_2_	**0.05 (0.01, 0.08)**	**0.008**
	RH: Medium NO_2_	**0.03 (0.00, 0.07)**	**0.050**

^1^ *p*-value < 0.05 in bold. ^2^ per 5% increase in RH at Lag01. * Adjusted for indicated study location, season, chronic respiratory disease, and indicated temperature.

## Data Availability

Requests for data should be addressed to the corresponding author.

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
