# Peer review of "The Influence of Short-Term Weather Parameters and Air Pollution on Adolescent Airway Inflammation"

_ijerph, 2023, doi:10.3390/ijerph20196827_

Round 1

Reviewer 1 Report

This manuscript entitled “The influence of short-term weather and air pollution on adolescent airway inflammation” mainly investigates the effect of weather parameters and air pollution on adolescents’ airway inflammation. The valuable data are helpful and vital in understanding the effect of meteorology and pollution on adolescents’ airways. On the other hand, the introduction and statistical analysis parts of the manuscript should be improved. I would suggest that the authors sincerely consider the following comments in the review/reconsider the valuable data and discussion.

Authors should use superscript for numbers throughout the manuscript for “PM2.5, O3 and NO2”

The title should be improved “The influence of short-term weather and air pollution on adolescent airway inflammation.” What do you mean? Short-term weather conditions/events/parameters?

Relevant previous studies should be given in the introduction part, especially for “ Although the short-term dose-response relationship between air pollution and FeNO has been described in children, young and older adults, there are limited data for adolescents” and “There is limited research on the interactive effect of weather and air pollution on FeNO”  and, their findings should be summarized. The introduction part should be improved.

Please add the Which analysis was applied to the data set, just “ linear correlation”? Which statistical package (R version 4.0.4? If yes, please add the statistical part or move) or which model was used for this study? Please improve the statistical analysis part.

What is Lag01 or Lag01 effect? It would be better to add an explanation for the readers.

Why did you try just for 5% in RH? Why just increase the RH?

“However, there were different sources of air pollution, with high traffic-related emissions likely explaining the higher concentrations of NO2 in Munich.” You can support your comment with the number of vehicles in the traffic.

Relevant previous studies should be given in the introduction part, especially for “ Although the short-term dose-response relationship between air pollution and FeNO has been described in children, young and older adults, there are limited data for adolescents” and “There is limited research on the interactive effect of weather and air pollution on FeNO”  and, their findings should be summarized. The introduction part should be improved.

Please add the Which analysis was applied to the data set, just “ linear correlation”? Which statistical package (R version 4.0.4? If yes, please add the statistical part or move) or which model was used for this study? Please improve the statistical analysis part.

What is Lag01 or Lag01 effect? It would be better to add an explanation for the readers.

Why did you try just for 5% in RH? Why just increase the RH?

“However, there were different sources of air pollution, with high traffic-related emissions likely explaining the higher concentrations of NO2 in Munich.” You can support your comment with the number of vehicles in the traffic.

Author Response

We thank the reviewers for their time in providing comments that helped to improve our manuscript. We resolved the specific comments as indicated below.

Please note: all changes/ revisions to the document are with track changes, including the abstract.

1. Authors should use superscript for numbers throughout the manuscript for “PM2.5, O3 and NO2”

We have made the changes requested throughout the paper.

2. The title should be improved “The influence of short-term weather and air pollution on adolescent airway inflammation.” What do you mean? Short-term weather conditions/events/parameters?

We thank the reviewer for this comment and have improved the title as suggested, it now reads, “The influence of short-term weather parameters and air pollution on adolescent airway inflammation.”

3. Relevant previous studies should be given in the introduction part, especially for “ Although the short-term dose-response relationship between air pollution and FeNO has been described in children, young and older adults, there are limited data for adolescents” and “There is limited research on the interactive effect of weather and air pollution on FeNO”  and, their findings should be summarized. The introduction part should be improved.

We have now provided relevant previous studies in the introduction and have summarised their findings. This can be found from Lines 58 to Line 70.

4. Please add the Which analysis was applied to the data set, just “ linear correlation”? Which statistical package (R version 4.0.4? If yes, please add the statistical part or move) or which model was used for this study? Please improve the statistical analysis part.

We apologise that this was unclear, we have now clarified in Line 133-134 that we performed linear regressions. In lines 160-164 we clarify the R packages we used.

5. What is Lag01 or Lag01 effect? It would be better to add an explanation for the readers.

We have clarified in Line 156 that Lag01 is a one-day moving average.

6. Why did you try just for 5% in RH? Why just increase the RH? 

Typically, weather parameter effects are calculated as per 1-unit/5-unit/10-unit increase in the variable. We chose 5% as it better describes a short-term effect.

7. “However, there were different sources of air pollution, with high traffic-related emissions likely explaining the higher concentrations of NO2 in Munich.” You can support your comment with the number of vehicles in the traffic.

It is not possible to give the number of vehicles in the traffic for Munich. Therefore, we clarified that Munich is Urban while Wesel is Rural.

Reviewer 2 Report

FeNO is recognized as biomarker of airway inflammation. Numerous studies investigated the relationship between air pollution, which is known to be the primary source of inflammatory processes in the airways and the lung, and the exhaled nitric oxide fraction (FeNO). Elevated FeNO has been connected to both short- and long-term exposure to air pollution at the population level.

In the present study, the authors investigated the main effects and interactions of low-level short-term air pollutants and weather parameters on adolescent’s FeNO. To reach this goal, 2042 participants from the German GINIplus and LISA birth cohorts' 15-year follow-up were included. Daily air pollution (ozone [O3], nitrogen dioxide (NO2), and particulate matter 2.5 m (PM2.5)) and meteorological data (maximum [Tmax], minimum [Tmin], mean [Tmean] temperatures, and RH) were evaluated. The authors searched for collinearity between variables and ran correlation tests as part of their statistical analysis. Ln(FeNO) was used as the outcome and continuous RH as the primary exposure in linear models that were fitted.

The several results obtained supported the idea that an increase in FeNO levels is associated with increased lung inflammation, which is linked to the interactions between RH and air pollution. In other words, higher FeNO was correlated with higher temperatures, PM2.5, O3, and NO2 levels.

The results presented in the present paper are interesting and important and might contribute to highlight the effect of the acceleration of climate change on human health.

I just recommend to the authors to elaborate further in the Introduction and Conclusion Sections.

Author Response

We thank the reviewers for their time in providing comments that helped to improve our manuscript. We resolved the specific comments as indicated below.

Please note: all changes/ revisions to the document are with track changes, including the abstract.

1. I just recommend to the authors to elaborate further in the Introduction and Conclusion Sections.

We thank the reviewer for this suggestion. We have elaborated and improved the Introduction by adding information that can be found from line 58 to line 71, and in the Conclusion section from line 374 to line 384.

Reviewer 3 Report

This interesting study relates the  Fraction of exhaled Nitric Oxide (FeNO) to the effects and interactions of relative humidity (RH) and air pollution on adolescents.

The following changes should be made.

1) In Tables 1 and 2, please compute a test to compare quantitative variables between both groups. Use Student’s T or the no parametric MannWitney’s U test to compare quantitative variables, depending on the variables' normal distribution.

2) In Tables 1 and 2, please compute a test to compare categorical variables between the 2 groups. Use the chi-square test  to compare categorical variables.

3) You should add a column with the P value of the comparisons in both tables.

Author Response

We thank the reviewers for their time in providing comments that helped to improve our manuscript. We resolved the specific comments as indicated below.

Please note: all changes/ revisions to the document are with track changes, including the abstract.

1. In Tables 1 and 2, please compute a test to compare quantitative variables between both groups. Use Student’s T or the no parametric MannWitney’s U test to compare quantitative variables, depending on the variables' normal distribution.

We have considered the reviewer's suggestion and have used the Wilcoxon rank sum test in both Table 1 and 2.

2. In Tables 1 and 2, please compute a test to compare categorical variables between the 2 groups. Use the chi-square test  to compare categorical variables.

We agree with the reviewer’s suggestion and have performed the chi-square test in both Tables 1 and 2.

3. You should add a column with the P value of the comparisons in both tables.

We have added a p-value column to both Tables 1 and 2.

Round 2

Reviewer 1 Report

The necessary corrections were made; the revised manuscript can be accepted in its present form for publication.